# A Pragmatic Bilayer Selective Emitter for Efficient Radiative Cooling under Direct Sunlight

**DOI:** 10.3390/ma12081208

**Published:** 2019-04-12

**Authors:** Yiwei Liu, Anqi Bai, Zhenggang Fang, Yaru Ni, Chunhua Lu, Zhongzi Xu

**Affiliations:** 1State Key Laboratory of Materials-Oriented Chemical Engineering, College of Materials Science and Engineering, Nanjing Tech University, Nanjing 210009, China; 2452615070@njtech.edu.cn (Y.L.); aqbai@njtech.edu.cn (A.B.); zzxu@njtech.edu.cn (Z.X.); 2Jiangsu Collaborative Innovation Center for Advanced Inorganic Function Composites, Nanjing Tech University, Nanjing 210009, China; 3Jiangsu National Synergetic Innovation Center for Advanced Materials (SICAM), Nanjing Tech University, Nanjing 210009, China; 4Key Laboratory of MEMS of Ministry of Education, Southeast University, Nanjing 210096, China

**Keywords:** radiative cooling, bilayer selective emitters, thermal radiation, functional thin films

## Abstract

Radiative cooling can make the selective emitter cool below ambient temperature without any external energy. Recent advances in photonic crystal and metamaterial technology made a high-efficiency selective emitter achievable by precisely controlling the emitter’s Infrared emission spectrum. However, the high cost of the photonic crystals and meta-materials limit their application. Herein, an efficient bilayer selective emitter is prepared based on the molecular vibrations of functional nanoparticles. By optimizing the volume fraction of the functional nanoparticles, the bilayer selective emitter can theoretically cool 36.7 °C and 25.5 °C below the ambient temperature in the nighttime and daytime, respectively. Such an efficient cooling performance is comparable with the published photonic crystal and metamaterial selective emitters. The rooftop measurements show that the bilayer selective emitter is effective in the ambient air even under direct sunlight. The relatively low cost and excellent cooling performance enable the bilayer selective emitter to have great potential for a practical purpose.

## 1. Introduction

Cooling, as one of the major end-uses of electricity, triggers massive amounts of energy consumption. According to the research conducted by the US Department of Energy, 15% of the electricity, which is consumed by American buildings, is used for operating air conditioners [1]. The passive radiative cooling can make the temperature of the radiators‘ surface decrease below ambient temperature without external energy. Therefore, radiaive cooling has the potential to lower the emissions of greenhouse gases as well as optimize the existing structure of the energy source [2]. The efficient radiative cooling devices are expected to have near unity infrared (IR) emissivity within 8 to 14 μm, which is known as the atmospheric window, in order to transfer the heat directly to outer space [3,4,5,6]. The significant radiative cooling at night by the selective emitters was achieved in previous works [7,8,9]. The daytime radiative cooling, however, is still a great challenge since solar energy is absorbed intensively by these emitters, which generates massive heat on the surface of the emitters [2,10]. Therefore, the effecient daytime cooling process requires selective emitters to have near unity solar reflectivity while emitting selectively and significantly within 8–14 μm in the IR region.

In recent years, the developments of photonic crystals and metamaterials enable the control of the material spectrum. Aaswath P. Raman et al. fabricated a photonic crystal that contains 7 alternating layers of HfO_2_ and SiO_2_ with varying thickness [2]. The cooling power of 40.1 W·m^−2^ and the temperature gap of 4.9 K between the photonic crystal and the ambient temperature were achieved experimentally [2]. Md Muntasir Hossain et al. achieved the day-time radiative cooling through micro conical arrays, which are made of an alternating Ge and Al layer on the Si plate [6]. This anisotropic metamaterial can theoretically cool more than 10 °C below ambient temperature [6]. Eden Rephaeli et al. designed a SiC/SiO_2_ double layer 2D photonic crystal. The cooling power of more than 100 W·m^−2^ in the daytime at the ambient temperature was achieved [11]. However, the high cost of the photonic crystals and metamaterials limits their application [12]. The material selective emitters were widely studied in previous works. C. G. Granqvist et et al. deposited the SiO onto the Al sheet by the evaporating deposition method to form the 0.1 μm thickness of the functional coating [13]. A sharp emission peak at 10 μm was achieved due to the lattice vibrational absorption of SiO [13]. However, the sharp emission peak cannot match well with the atmospheric window, which limits its cooling performance [10]. Angus R. Gentle et al. prepared a SiC/SiO multi-layer thin film. This functional thin film reaches near unity IR emissivity within the atmospheric window because of the surface phonon resonance of SiC and SiO [14]. The thin film has, however, significant IR emission outside the atmospheric window, which leads to the poor selectivity of IR emission and the limited cooling performance.

Herein, we designed a bilayer selective emitter. On the top of the emitter is Polymethylpentene (TPX) thin film embedded with SiO_2_ and CaMoO_4_ functional nanoparticles as a selective IR emitting layer. The IR emission of the functional thin film can match well with an atmospheric window due to the O–Si–O asymmetric vibrations of SiO_2_ (8–10 μm) [15] and the Mo-O stretching mode of CaMoO_4_ (11–14 μm) [16]. The bottom layer of the selective emitter is the 600 nm Ag, which was deposited on the Si sheet, as the solar reflecting layer. The functional TPX thin films with different nanoparticle volume fractions and film thickness were prepared to optimize the IR emission performance. The optimized bilayer selective emitter exhibits strong and strictly selective IR emission within 8–14 μm, which enables it to cool 36.7 °C and 25.5 °C below the ambient temperature at night and in the daytime under direct sunlight theoretically. To evaluate the cooling performance in the real environment, the rooftop measurements were carried out and the effective cooling performance was achieved.

## 2. Materials and Methods

### 2.1. Preparation of Bilayer Selective Emitter

The preparation process of the bilayer selective emitter is shown in Figure 1. The CaMoO_4_ nanoparticle was prepared by the sol-gel method [17]. Stoichiometric amounts of Ca(NO_3_)_2_·4H_2_O (SCR Chemicals, Johannesburg, South Africa) and (NH_4_)_6_Mo_7_O_24_·_4_H_2_O (SCR Chemicals) were first dissolved in distilled water. Then the solution was added to 2.5 mol/L C_6_H_8_O_7_·H_2_O (Sinopharm Chemicals, Shanghai, China) solution and the NH_3_·H_2_O (Ling-Feng Chemicals, Shanghai, China) was used to adjust the pH value to 3. Then, the solution was stirred at 75 °C for 12 h to form a transparent gel, and the transparent gel was heated at 120 °C for 3 h to produce the solid precursor. The precursor was then calcined at 850 °C in air for 3 h. 

To produce the functional thin film, 1.25 g TPX raw material (Mitsui Chemicals, Tokyo, Japan) was dissolved in 40 mL trichloroethylene (Ling-Feng Chemicals, Shanghai, China) at 50 °C. Then, the CaMoO_4_ and the SiO_2_ (Aladdin Chemicals, Shanghai, China) nanoparticles were added to the solution. After 12 h of ball-milling, the nanoparticles were randomly dispersed into the TPX solution. Lastly, the disperse system was used to produce the functional thin film on the surface of the Ag layer through a coating machine (OSP Mechanical Technology Co., Shijiazhuang, China). The Ag layer was previously deposited on the Si substrate using the Physical Vapor Deposition method (PVD, Guotai Vacuum Equipment Co., Chengdu, China).

### 2.2. Characterization Methods

The phase composition of the nanoparticles was analyzed by an X-ray diffractometer (XRD, Rigaku Smart Lab, Tokyo, Japan) with Cu Kα radiation. The working voltage and current are 35 KV and 30 mA, respectively. The scans were performed in the range of 10° to 80° with an angle step of 0.02°. Particle size distributions of the nanoparticles are measured by Particle Sizer and Zeta Potential Analyzer (Brookhaven Instruments Corp., New York, NY, USA). The nanoparticles were dispersed in ethyl alcohol before the measurement. The thickness of the functional thin film of these samples was measured by the coating thickness gauge (Fisher MPO, Bad Salzuflen, Germany). We measured the solar and IR reflectivity R(λ) using the UV-VIS-NIR Spectrophotometer (SHIMADZU, Kyoto, Japan) and FTIR spectrometer (PerkinElmer, Waltham, MA, USA). Because the emitter is opaque to visible and infrared light and E(λ) = a(λ), where E(λ) and a(λ) are emissivity and absorptivity, according to Kirchhoff’s law of thermal radiation, E(λ)/a(λ) can be calculated by 1—R(λ). 

The rooftop measurements were carried out and lasted for 24 h on 12 March 2019 in Nanjing, China, with relative air humidity of 53%. The bilayer selective emitter was placed in an expanded polystyrene (EPS) foam box to reduce heat conduction between the selective emitter and the ambient temperature. A high solar reflectivity aluminum sheet to minimize the influence of the test set covered the internal and external surfaces of the boxes. A clear low-density polyethylene (LDPE) was covered on the test setup so nonradiative heat exchange can be reduced and a relatively well-sealed air pocket was created for the emitter. For comparison, two more test boxes were prepared in which the polished Si substrate and the Si substrate was deposited on 600 nm Ag, respectively. All the setups were tilted 30° toward the south so solar irradiance can reach the surface of the selective emitter vertically. This can maximize the solar radiative power input while reducing sky access for thermal radiation [2]. Therefore, if another tilt was chosen, better performance can be expected. The back of the samples was loaded with temperature sensors that were connected to the data logger. All the sensors were calibrated previously so the measured temperature differences among these sensors are less than 0.2 °C. In addition, the same sensor was placed in a sun-shaded area with free airflow near but outside the boxes to measure the ambient temperature. The solar irradiance was measured by the optical power density meter (Newport Power Meter). All the data were recorded every minute.

## 3. Results

### 3.1. The Structure of the Bilayer Selective Emitter

The selective emitter has a bilayer structure, as shown in Figure 2a, on the bottom of which is 600 nm thickness of Ag deposited on polished Si substrate as the solar reflecting layer. The upper layer is the TPX thin film embedded randomly with SiO_2_ (Aladdin Chemicals) and CaMoO_4_. TPX provides an excellent solar and IR transmittance [18]. Figure 2b shows the X-ray diffraction patterns of the as-synthesized CaMoO_4_ nanoparticles. As can be seen from the XRD patterns, all diffraction peaks can be identified as CaMoO_4_ (Pdf No. 7-212) phases and no impurity phase was detected, which indicates that the single phase of CaMoO_4_ was obtained (as confirmed by the energy-dispersive X-ray spectroscopy (EDS) spectra in Figure 2c). The XRD pattern of the SiO_2_ nanoparticle (as confirmed by the EDS spectra in Figure 2d) indicates that the SiO_2_ is amorphous. The particle size distribution of the nanoparticles shown in Figure 2b indicate that the mean diameter of SiO_2_ and CaMoO_4_ are 409.4 nm and 433.2 nm, respectively. The Transmission Electron Microscope (TEM) image of the CaMoO_4_ nanoparticle shown in Figure 2e indicates that the distance of its lattice strips is 3.09 Å, which corresponds to the (112) plane of the tetragonal phase CaMoO_4_. TEM image of the SiO_2_ nanoparticle shown in Figure 2f further demonstrates its amorphous structure since no trace of any lattice strip can be identified. Figure 2g shows the Scanning Electron Microscope (SEM) image of the cross morphology of the bilayer selective emitter from which the 600 nm Ag layer on the surface of the Si substrate is clearly identified. The TPX functional film covers the Ag layer. We produced a circular bilayer selective emitter with a diameter of 15 mm for the spectrum analysis and the rooftop measurements.

### 3.2. Optical Characteristic

To better analyze the effects of different components on the optical property, we prepared samples embedded with a 15% SiO_2_ nanoparticle by volume (TPX + 15% SiO_2_) and a TPX thin film embedded with a 15% CaMoO_4_ nanoparticle by volume (TPX + 15% CaMoO_4_), respectively. As shown in Figure 3a, a low solar absorptivity is observed for all the samples. As shown in Figure 3b, the TPX film shows excellent IR transmittance. The samples containing 15% SiO_2_ and 15% CaMoO_4_ nanoparticles can emit effectively within 8 to 10 μm and 11 to 14 μm, respectively, while having weak IR emission beyond the atmospheric window.

We prepared samples with a different nanoparticle volume fraction to optimize the emitter’s IR emission characteristic. The samples and their corresponding particle volume fraction are shown in Table 1. $T_{a}$ and $T_{r}$ in Table 1 is the ambient and the emitter temperature, respectively. Figure 4a demonstrates the measured solar absorptivity of these samples, which is necessary for the evaluation of the samples’ cooling performance. The results show that all samples have limited solar absorption due to the high solar reflectivity of the Ag layer and limited solar absorptivity of the functional TPX thin film, which is essential to achieve effective day-time cooling. As shown in Figure 4b, when the nanoparticle volume fraction increases, the IR emissions within and outside the atmospheric window are both enhanced. The Solar irradiance I_AM_1.5(λ) is ASTM G173-03 Reference Spectra derived from SMARTS v. 2.9.2 (AM1.5) and the atmospheric transmittance t(λ) is calculated using MODTRAN 5 with a relative humidity of 60% [19,20].

Samples with different film thickness are also prepared. All the samples have the same nanoparticle volume fraction (15% SiO_2_ and 15% CaMoO_4_). As shown in Figure 4c, all of these samples exhibit limited solar absorptivity. Figure 4d shows that, when the thickness of the film increases, the IR emissions within and outside the atmospheric window are both increased.

### 3.3. Cooling Performance Evaluation

To numerically identify the optimal volume fraction of the functional nanoparticles and film thickness among these samples, we evaluated their theoretical cooling performance by a model shown as follows. Considering all heat change processes exist on a surface, the net cooling power of a radiative cooler P_net_ can be calculated by the equation below.
(1)$$P_{\mathrm{net}}{=P}_{r}-P_{a}{-P}_{\mathrm{nonrad}}{-P}_{\mathrm{sun}}$$
where
(2)$$P_{r}=2\pi \overset{\pi /2}{\underset{0}{\int }}\sin\theta \cos\theta d\theta \overset{\infty }{\underset{0}{\int }}B\left(T_{r},\lambda \right)e_{r}\left(\lambda ,\theta \right)d\lambda$$
is the radiative power density emitted by the radiator.
(3)$$P_{a}=2\pi \overset{\pi /2}{\underset{0}{\int }}\sin\theta \cos\theta d\theta \overset{\infty }{\underset{0}{\int }}B\left(T_{a},\lambda \right)e_{a}\left(\lambda ,\theta \right)e_{r}\left(\lambda ,\theta \right)d\lambda$$
is the incident atmospheric radiation absorbed by the radiator.
(4)$$P_{\mathrm{nonrad}}=q\left(T_{a}{-T}_{r}\right)$$
is nonradiative heating power obtained by the radiator from the surrounding media.
(5)$$P_{\mathrm{sun}}=\overset{\infty }{\underset{0}{\int }}e_{r}\left({\lambda ,\theta }_{\mathrm{sun}}\right)I_{\mathrm{AM}1.5}(\lambda )d\lambda$$
is the solar power absorbed by the radiator.
(6)$$B\left(T,\lambda \right)=\frac{{2\mathrm{hC}}^{2}}{\lambda^{5}}\times \frac{1}{e^{\frac{\mathrm{hC}}{\lambda \mathrm{kT}}}}-1$$
is the spectral radiance of the black body at the temperature T according to Planck’s law. C, K, and h represent the speed of light, the Boltzmann constant, and the Planck constant, respectively. The emissivity of the radiator, according to Kirchhoff’s law, is equal to its absorptivity $e_{r}\left(\lambda ,\theta \right)$. $e_{a}\left(\lambda ,\theta \right)$ in Equation (3) represents angle dependent emissivity of the atmosphere, which can be defined by $e_{a}\left(\lambda ,\theta \right){=1-t(\lambda )}^{1/\cos\theta }$ [2]. In Equation (4), the combined nonradiative heat coefficient q can be defined by ${q=q}_{\mathrm{conduct}}{+q}_{\mathrm{convection}}$, where $q_{\mathrm{conduct}}$ and $q_{\mathrm{convection}}$ represents the effect of conductive and convective heat coefficient, respectively, which are subject to the ambient temperature in which the radiator is located. q varies from 0 to 6.9 W·m^−2^·K^−1^ with the change of environment and testing device [2,6,10,12]. $e_{r}\left({\lambda ,\theta }_{\mathrm{sun}}\right)$ in Equation (5) is the absorptivity of the radiator relating to the solar incident angle $\theta_{\mathrm{sun}}$. Equation (1) illustrates the daytime cooling power of a radiator. If a positive $P_{\mathrm{net}}$ is achieved in the initial state ($T_{a}{=T}_{r}$), the radiator can be defined by a daytime cooling device. Furthermore, the temperature difference $T_{a}{-T}_{r}$ is expected to reach a steady state when $P_{\mathrm{net}}=0$, which means there is no extra power for the radiator to further cool down. Thus, the value of the temperature difference $T_{a}{-T}_{r}$ in a steady state can be used to evaluate the performance of a selective radiator quantitatively.

Table 1 shows the steady state temperature difference $T_{a}{-T}_{r}$ of the samples with different nanoparticle volume fractions under direct sunlight (AM 1.5). The ambient temperature was set to 300.15 K and both the conductive and convective heat transfer were excluded (q=0 W·m^−2^·K^−1^). As can be seen, $T_{a}{-T}_{r}$ increases with the rising nanoparticle volume fraction when the volume fraction is relatively low. The negative temperature of sample S-5 means that it can reach 3.2 °C higher than the ambient temperature under the direct sunlight when it maintains the steady state due to the weak IR emission within 8 to 14 μm (Figure 5b). However, with the volume fraction further increasing, $T_{a}{-T}_{r}$ decreases as the result of degraded emission selectivity. When the sample with the highest $T_{a}{-T}_{r}$ was at a steady state, S-15 was selected for further investigation. Table 2 shows the $T_{a}{-T}_{r}$ of the samples with different film thicknesses. As can be seen, for the samples that contain 15% (volume fraction) SiO_2_ and 15% (volume fraction) CaMoO_4_, the optimal thickness is 3 μm since the sample with 3-μm exhibited better cooling performance.

Figure 5a shows the theoretical nighttime (without solar irradiation) cooling performance of sample S-15 with the film thickness of 3 μm. As can be seen, an initial state net cooling power of 96.4 W·m^−2^ is achieved. In addition, the bilayer selective emitter can significantly cool 36.7 °C below the ambient temperature (27 °C) with the absence of nonradiative heat exchange. When q increases from 0 to 6.9 W·m^−2^·K^−1^, the gap between the emitter temperature and ambient temperature at a steady state decreases from 36.7 °C to 19.8 °C. As shown in Figure 5b, for cooling performance under direct sun light (AM 1.5), a temperature difference of 16.6 °C is obtained when q=0 W·m^−2^·K^−1^. It is noteworthy that the bilayer selective emitter can still cool 4.9 °C below the ambient temperature even when q=6.9 W·m^−2^·K^−1^ because of its fairly low solar absorption and significant selective emission within the ‘atmospheric window.’ To evaluate the emitter’s cooling performance in different ambient conditions, we simulated the atmospheric transmittivity in different relative humidities using MODTRAN 5. The results shown in Figure 5c inllustrate that the IR transimitivity of atmosphere decreases with the increment of the relative humidity of the ambient temperature. The corresponding theoretical day-time cooling performance of the emitter are shown in Figure 5d, from which a cooling performance of 25.5 °C in relatively dry ambient air can be identified. When compared to other material selective emitter, our bilayer selective emitter achieves better cooling performance. The comparison between the bilayer selective emitter and a selective emitter in Reference [18] (Appendix A) indicates that the bilayer selective emitter has better theoretical cooling performance. A. R. Gentle and G. B. Smith demonstrated a radiative cooling thin film that theoretically cools 40 °C below ambient temperature at night with q=0 W·m^−2^·K^−1^ [21]. It should be mentioned that the coating produced by A. R. Gentle and G. B. Smith is designed only for nighttime cooling. The bilayer selective emitter, however, can cool significantly at night while achieving efficient daytime cooling. In terms of theoretical daytime cooling performance, the bilayer selective emitter is comparable with a published photonic crystal and metamaterial selective emitters [2,6,11,22]. The relatively low cost enables the bilayer selective emitter to have great potential for a practical purpose.

To evaluate the cooling performance of the bilayer selective emitter in a real environment, the rooftop measurements were carried out. The pictures and schematic images of the test apparatus are shown in Figure 6a,b, respectively. Figure 6c demonstrates the results of the rooftop measurement. It is shown that only the bilayer selective emitter can have the temperature that is lower than the ambient temperature while the other two samples’ temperatures reach higher than the ambient temperature under the direct sun light. Figure 6d demonstrates the temperature difference between the bilayer selective emitter and the ambient temperature shown in Figure 6c from which a temperature gap of about 5 °C in the nighttime was identified. Furthermore, the temperature of the emitter was lower than the ambient temperature even when the significant solar light irradiated on the sample in the daytime. It should be mentioned that the environmental temperature during the rooftop measurements is relatively low (as can be seen from Figure 6c). However, radiative cooling devices can have better cooling performance when the ambient temperture is relatively high. In addition, for most of the time, the radiative cooling devices are used in a high temperature environment.

## 4. Discussion

A bilayer selective emitter was prepared. Spectral analysis of the samples with a different volume fraction of the functional nanoparticle illustrates that the selectivity of the emitter’s IR emission reduces with the increase of the particle’s volume fraction, which can degrade the performance of the selective emitter. The theoretical cooling performance of these samples were theoretically analyzed. The results show that the sample contained 15% SiO_2_ and 15% CaMoO_4_ (volume fraction), which exhibits better cooling performance. The experimental results of the samples with different film thicknesses indicate that the optimal thickness for the sample that contained 15% SiO_2_ and 15% CaMoO_4_ (volume fraction) is 3 μm. The bilayer selective emitter with this functional nanoparticle content and film thickness can theoretically cool to 36.7 °C and 16.6 °C below the ambient temperature (27 °C) in the nighttime and under direct sun light (AM1.5), respectively, with the absence of nonradiative heat exchange (q=0 W·m^−2^·K^−1^) and relative humidity at 60%. Furthermore, a daytime cooling performance of 25.5 °C during the daytime (AM1.5) was identified when the ambient air was relatively dry. To the best of our knowledge, the cooling performance of the bilayer selective emitter is better than that of other published material selective emitters and is comparable with the published photonic crystal and metamaterial selective emitters. The relatively low cost and excellent cooling performance enable the bilayer selective emitter to have great potential for a practical purpose.

## Figures and Tables

**Figure 1 materials-12-01208-f001:**
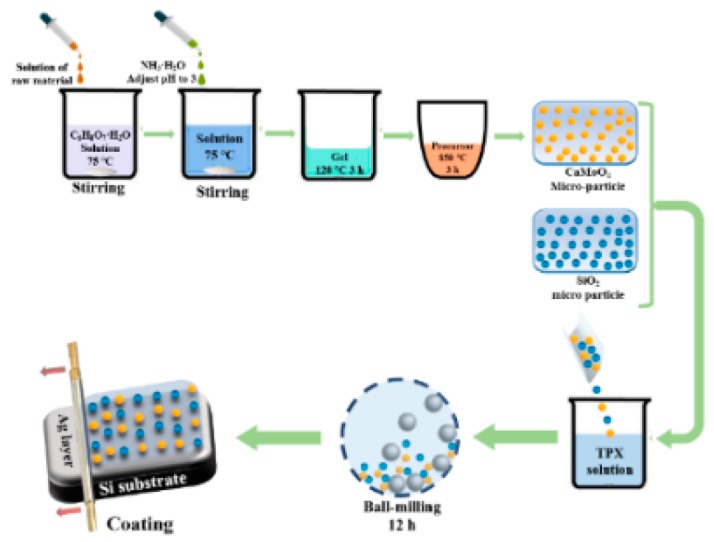
Schematic of the preparation process of the bilayer selective emitter.

**Figure 2 materials-12-01208-f002:**
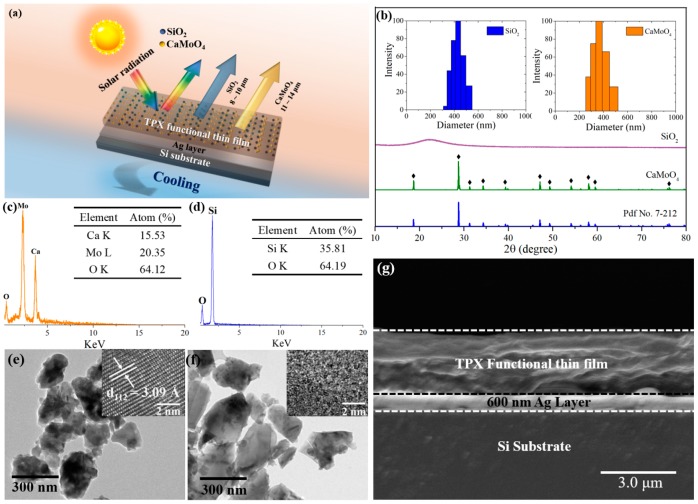
(**a**) Schematic drawing of the bilayer selective emitter. (**b**) X-ray diffraction patterns of the CaMoO_4_ and SiO_2_ nanoparticles. Insets are the particle size distribution of the nanoparticles. (**c**,**d**) Energy dispersive spectrometer (EDS) patterns of (**c**) CaMoO_4_ and (**d**) SiO_2_ nanoparticles. (**e**,**f**) TEM imagines of (**e**) CaMoO_4_, and (**f**) SiO_2_ nanoparticles. (**g**) SEM images of the section morphology of the emitter.

**Figure 3 materials-12-01208-f003:**
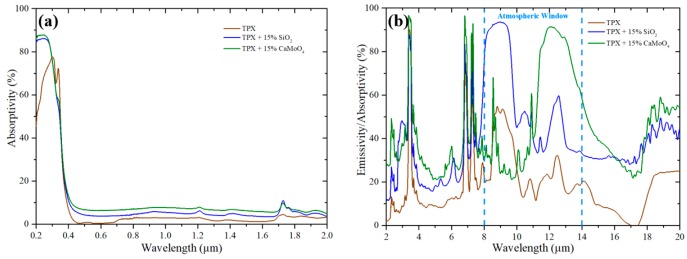
Measured emissivity/absorptivity spectra of the samples contained with a different functional nanoparticle. (**a**) Solar absorptivity of the samples. (**b**) IR emissivity/absorptivity of the samples.

**Figure 4 materials-12-01208-f004:**
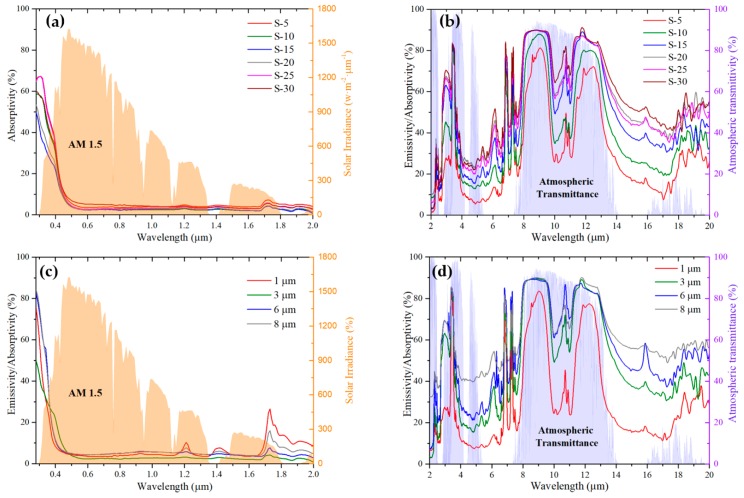
(**a**) Solar absorptivity and (**b**) IR emissivity/absorptivity of the samples with a different nanoparticle volume fraction. (**c**) Solar absorptivity and (**d**) IR emissivity/absorptivity of the samples with different film thicknesses.

**Figure 5 materials-12-01208-f005:**
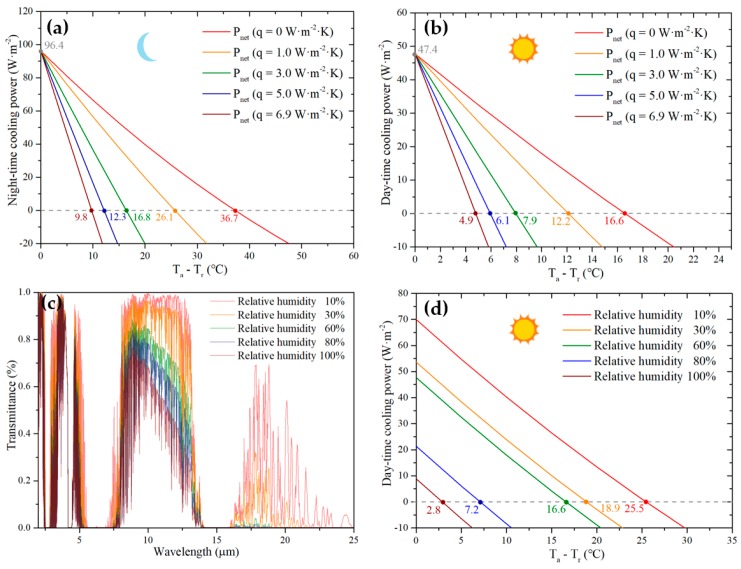
Theoretical (**a**) nighttime and (**b**) daytime net cooling power of the bilayer selective emitter with relative humidity 60%. (**c**) Atmospheric transmittivity with different relative humidity. (**d**) Daytime cooling performance of the emitter with different relative humidity.

**Figure 6 materials-12-01208-f006:**
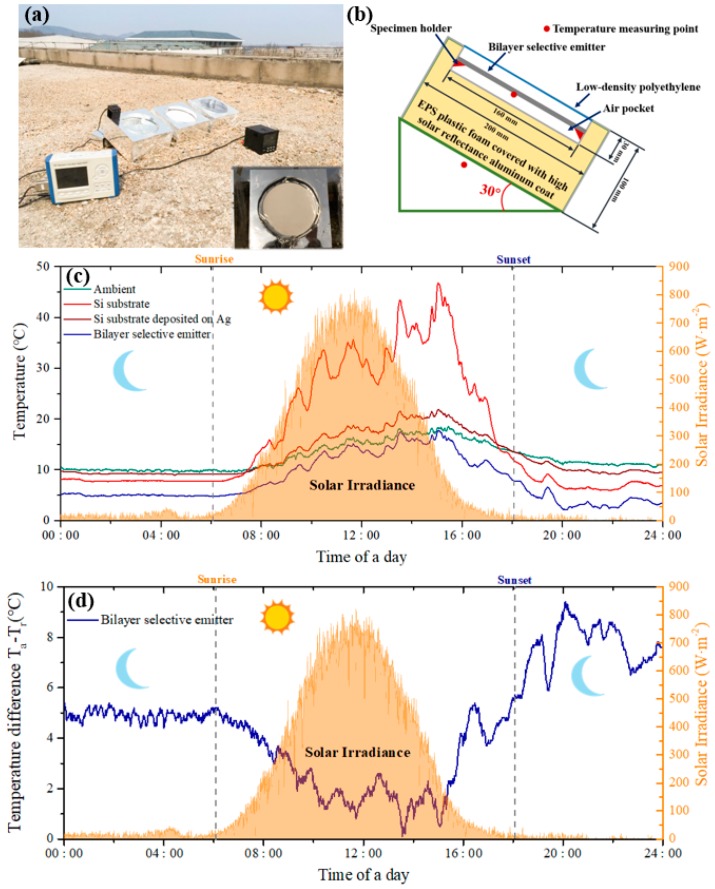
The rooftop measurement apparatus and the results. (**a**) The picture of the testing apparatus. The inset is the picture of a testing box in which the bilayer selective emitter is located. (**b**) The cross section schematic of the testing boxes. (**c**) A 24-h cooling test results of the rooftop measurement. (**d**) The temperature difference between the bilayer selective emitter and the ambient within 24 h.

**Table 1 materials-12-01208-t001:** Samples with different nanoparticle volume fraction and their corresponding theoretical cooling performance.

Samples	S-5	S-10	S-15	S-20	S-25	S-30
Volume fraction of SiO_2_	5%	10%	15%	20%	25%	30%
Volume fraction of CaMoO_4_	5%	10%	15%	20%	25%	30%
T_a_ − T_r_ (°C)	−3.2	10.4	16.6	14.9	11.1	7.5

**Table 2 materials-12-01208-t002:** Theoretical daytime cooling performance of the samples with different film thicknesses.

Sample Thickness	1 μm	3 μm	6 μm	8 μm
T_a_ − T_r_ (°C)	6.9	16.6	12.1	10.3

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
