# Peer review of "A Pragmatic Bilayer Selective Emitter for Efficient Radiative Cooling under Direct Sunlight"

_materials, 2019, doi:10.3390/ma12081208_

Round 1
Reviewer 1 Report
Review of ‘A pragmatic bilayer selective emitter for efficient radiative cooling under direct sunlight’
This paper presents experimental absorption measurement of their bilayer selective emitter and used it to model the radiative cooling performances. The structure is very similar to the published structure of Y. Zhai and Xiaobo Yin’s (ref. 17). The difference seems to be just embedding additional CaMnO4 microparticles. Nonetheless, it is an important contribution to the field if indeed the efficiency of the selective emitter is better. Therefore, I think should address 1) their result is directed compared with the ref. 17; 2) The role of CaMnO4 is discussed, before it can be published.
To elaborate on these requests; since the emissivity curve of ref. 17 was published, the authors should be able to plug their data into their model and compare the performance directly. I raise this question because the emissivity curves shown in fig. 5 has a hole in the 10-11um, whereas ref. 17 does not, hence the radiative cooling efficiency should suffer. This lead to another question which is why should there be a hole in the emissivity curve in the 10-11um region?
In addition, some editorial changes need to be make.
1. There are some editorial mistakes in the introduction paragraph, this should be edited by people with higher proficiency in English.
2. The ‘materials and methods’ part is also poorly written. It is not clear enough for reader to reproduce their recipe.
3. The axis labels of fig. 1b are too small.
4. For fig 4b, why is the emissivity of 15% SiO2 and 15% CaMnO4 in TPX not shown case since it is has the best performance shown in Table 1.
Reviewer 2 Report
The authors describe a method to create a radiative emitter out of SiO2 and CaMoO4 nanoparticles embedded in a polymer (PTX) thin film, which is placed on top of a silver back reflector. The purpose of this device is to achieve day time radiative cooling. The working principle is that the combination of SiO2 and CaMoO4 nanoparticles in PTX can provide over 80% absorptivity/emissivity in approximately 2/3 of the atmospheric window. The device is also over 90% reflective between 500nm – 2um. Proper control experiments were performed and quantify the characteristic reflectivity and emissivity of the structure as a function of nanoparticle volume fraction. Based on the control experiments calculations of theoretically ideal cooling performance was evaluated. No empirical cooling measurements were performed in either a lab or outdoor setting. Overall, I think the idea is logical, interesting, and a possibly simple method to create a radiative cooling structure. The use of CaMoO4 nanoparticles for daytime radiative cooling is not extensively studied and appears sensible as a mixture with SiO2 nanoparticles. SiO2 has been extensively studied for radiative cooling. With that being said, I feel the following should be addressed before consideration of publication:
1) Many radiative cooling designs show high theoretical cooling efficiency, but fall short in empirical measurement. It would be useful to perform a radiative cooling experiment to determine if the proposed structure can actually achieve daytime radiative cooling.
2) The manuscript has many consistent grammatical errors. At least one grammatical error was found in five out of the seven sentences composing the first paragraph. The primary grammatical errors were a lack of articles, the incorrect use of plural and singular nouns, and redundant words.
3) The optimal nanoparticle volume fraction for radiative cooling was reported to be 15%. It is unclear why this value is optimal. Since this conclusion comes from theoretical calculations, it would be useful if the authors could show how the cooling performance behaves as you deviate from the optimal volume fill fraction.
4) It would be useful to provide a scientific explanation or experiment detailing why the nanoparticle composite film thickness was chosen to be 3um. Film thickness can be an important design parameter for achieving daytime radiative cooling. It is currently unclear how the role of film thickness is contributing to cooling efficiency in this design.
5) The authors should consider characterizing the particles using energy-dispersive X-ray spectroscopy (EDS) alongside their XRD data to corroborate their characterization results. Trace amounts of crystalline impurities tend to be missed due to the size limitations of XRD. EDS would provide chemical composition data affirming the stoichiometry of the composition as well as suggest chemical impurities present in the particles.
The following minor corrections are also proposed:
1) Please change micro-particles to nanoparticles. The size distribution of the particles shows no particle size above 600nm.
2) Please define the symbolic mathematical variables Ta and Tr in section 3.2, where they are first used. The variables are not defined until section 3.3.
3) There is no citation for the synthesis of the CaMoO4 nanoparticles. Is this process unpublished? Please add a citation for the synthesis process if it is necessary.
Round 2
Reviewer 1 Report
Review of ‘A pragmatic bilayer selective emitter for efficient radiative cooling under direct sunlight’
It is nice that the authors have some experimental data.
The authors’ response to my first question on comparing Xiaobo’s work did not really address my comment. In the revised version of the manuscript, the conclusion still claims their structure is better than Xiaobo’s structure, but provided no supporting evidence. Yet, in the response letter, the authors claimed that they don’t have the raw data to make the comparison. These statements are contradictory. There are plenty of software out there that they can digitize their emissivity structure and make comparison.
2. The experimental data need to be reduced and compared with their own theory this is not done.
In my opinion this article should not be published unless proper comparison with their own theory and other published results.
Round 3
Reviewer 1 Report
Thank you for the comparison study with respect to ref. 17. Looking at the emissivity plots alone suggests bilayer should have a better performance. However, I do find the results in Fig 2 in the response document puzzling. Ref. 17 emitter has a larger cooling rate than the bilayer structure at zero temperature difference and yet yields a smaller temperature difference at the net zero cooling rate assuming both structures have the same parasitic heating of the sample. This seems counter intuitive.
Also, I don't understand why none of the comparison studies/observations is being presented in the revised manuscript.
For Fig 6, the difference temperature should be plotted.
